

# A strong link between variations in sea-ice extent and global atmospheric pressure ?

Jean-Louis Le Mouël[1], Fernando Lopes[1], Vincent Courtillot[1]

[1]Institut de Physique du Globe, Paris University, Paris 75005, France

*Correspondence to*: Vincent Courtillot (courtil@ipgp.fr)

**Abstract.** This paper reports spectral analyses, using Singular Spectral Analysis, of variations of the Arctic and Antarctic sea-ice extents (SI), and of the atmospheric surface pressure (AP) in both hemispheres (NH and SH). The ice-extents are dominated by a quasi-linear trend over the 42 yr period when data are available (1978-2020) and an annual component. Taken together, these two components represent more than 98% of the signal variance. Both ice-extent series

share the same set of harmonics of the annual component (1/2, 1/3, ¼ and 1/5 yr). The multi-decadal trends of sea-ice extent in the Arctic and Antarctic are of opposite sign. The series of harmonics of 1 year are consequences of the Earth's revolution about the Sun. The components with period longer than a year form a set of even harmonics of the Schwabe cycle. The pressure series also exhibits the four harmonics of 1 year, that is not found in many series previously analysed in the same way. This could suggest a connection between variations in pressure and sea-ice extent. Geographical pressure structures

(SSA trends) are stable on a decadal to centennial time scale and exhibit a three-fold symmetry in the NH. In the SH that order-3 symmetry is altered by the Ross-Weddell "dipole" pressure anomaly. This anomaly is seen in maps of correlations of variations in sea-ice extent with atmospheric pressure, surface temperature and winds. It fits topographic forcing. There is phase opposition between the annual components of SI and AP in the SH, and the same decreasing phase lag from -30 to -60 days over 42 years for the four harmonic components of SHSI and SHAP. The (negative) sign of the trend of pressure and

(positive) sign of the trend of temperature beg for an explanation. The relative change in pressure over the past 50 years is two orders of magnitude smaller than that of warming. This relatively strong warming would be expected to have a larger effect on pressure. The ratio of relative changes of sea-ice extent vs pressure is 400 for the NH and 17 for the SH. The SSA components reported in this paper should help in understanding the mechanisms that govern changes in sea-ice extent: these changes reflect forcings related to the Earth's revolution about the Sun on the shorter period side, and on the longer period

side to the Sun and planets (Jupiter). Advanced explanation of the physics underling these observations may need advances in solving the generalized Navier-Stokes equations, which is very difficult in the spherical case.

## 1 Introduction

Ice, and in particular higher latitude sea ice (SI), is an important component of the climate system. Passive-microwave

observations from satellites provide the best data yet available for observing the full Arctic and Antarctic ice covers on a repetitive basis throughout the year (e.g. Cavalieri and Parkinson, 2012). These data are available since November 1978. In





the present paper, we use singular spectrum analysis (SSA, see Golyandina and Zhigljavsky, 2013) algorithms to identify components of the variability of sea ice cover in the higher latitudes of both the Arctic (extent of the Northern hemisphere sea ice, NHSI) and Antarctic (extent of the Southern hemisphere sea ice, SHSI). A short description of SSA is given in

Lopes et al. (2017) and Le Mouël et al. (2020b). We compare our results to those obtained in previous studies, notably Parkinson et al (1999) and Cavalieri and Parkinson (2012) for the Arctic and Bertler et al. (2018) and Parkinson (2019) for the Antarctic.

There is a strong dissymmetry between variations in ice extent in both hemispheres (and also accessibility). As a result, there are far more studies and papers on NH than SH sea-ice. In the NH there is a long-term (multi-decadal) decrease

in the surface area of SI, which is in general attributed to anthropogenic warming due to increased atmospheric $CO_2$ (*e.g.*, quoting only papers published in the past few years, Notz  et al., 2020; Park and Kug, 2020; Dai et al., 2019; Kwok, 2018; Stroeve and Notz, 2018). It may seem puzzling that in the SH the trend is opposite (*e.g.* Parkinson, 2019).

Analyses are based on the identifications of sub-zones of the Antarctic where the trend of sea-ice cover variations can either be an increase or a decrease, and one seeks to understand the cause for such geographical patterns (Parkinson, 2019).

In some areas where one might expect melting, it is actually ice growth that prevails (*e.g.* De Santis et al., 2017; Pope et al., 2017; Kim et al., 2018; Farooq et al., 2020; Turner et al., 2020)

It has often been considered since the 1990s that climate variations in the NH and SH are forced or at least modulated by large-scale atmospheric oscillations, whose indices, AO for the Arctic (Thompson et Wallace, 1998) and AAO for the Antarctic (Gong et Whang, 1999), are coupled (Guan and Yamagata, 2001; Lu et al., 2008; Guan et al., 2010; Tachibana et

al., 2018). When attempting to compare the Arctic and Antarctic based on these indices, one should keep in mind that AAO is built in a rather straightforward way from pressures at sea level between 40°S and 65°S latitudes, whereas AO is built in a much more complex way that includes temperatures at sea level. More precisely, environmental variability in the southern (polar) regions is dominated by the interplay of three oscillations, the Southern Annular Mode (SAM), El Nino-Southern Oscillation (ENSO) and Inter Decadal Pacific Oscillation (IPO) (Bertler et al., 2018).

In earlier work (Le Mouël et al., 2019a), we have shown (using SSA) that the same pseudo-cycles, essentially linked to (and only to) solar activity, affect most if not all climate indices. For instance, our analysis of the AAO index revealed cycles with periods 11, 5.5 and 3.3 years, that can be attributed to the Schwabe cycle and its first two harmonics (within uncertainty) (Le Mouël et al., 2020a; Courtillot et al., 2021). These cycles are present in surface temperatures (Le Mouël et al., 2020b) and also in the magnetic field (Le Mouël et al., 2019b) and motion of the pole (Lopes et al., 2017; Lopes et al.,

2021). These oscillations, connected to solar activity, imply a reorganization of fluid masses at the Earth's surface at the same periods, ranging from several years to several decades (Le Mouël et al., 2021, Lopes et al., submitted). In the period range of one year and less, shorter pseudo-periods are identified (between 6 months and 13.6 days, in the Madden-Julian (MJO) index (Le Mouël et al., 2019a) and also in the length of day lod (Le Mouël et al., 2019c).



In the present paper, we wish to investigate whether similar oscillations can be found in the NHSI and SHSI time series, and which similarities and differences between the two hemispheres might be found. We explore in particular the potential links between atmospheric sea-level pressure and sea-ice extent.

We have studied with SSA the variations and oscillatory components of many solar and climatological and geophysical phenomena (see the above references and more references in these papers), but atmospheric pressure (AP). Yet, pressure is important in accounting for winds, and winds are an important component of the growth and disintegration of sea-ice. So, we also include in the present paper an SSA analysis of atmospheric pressure above polar regions. We describe the data and methods in section 2, and present the main results for the shorter periods in section 3 (complemented by an Appendix for the longer periods). We explore the case of Antarctica in more detail in section 4 and discuss possible interpretations and consequences of our findings in section 5.

## 2 Data and Method

In this section, we explain where the data for NHSI, SHSI and AP can be obtained.

The data set (Cavalieri and Parkinson, 2012; Fetterer et al, 2017) from which the *Arctic (NHSI) and Antarctic (SHSI) sea-ice areas* are calculated consists of sea ice concentration maps derived from the radiances obtained from a suite of satellite microwave radiometers. This started with the Nimbus 7 Scanning Multichannel Microwave Radiometer (SMMR), which operated from 1978 through 1987, then the Defense Meteorological Satellite Program (DMSP) series of F8, F11, F13 and F15 Special Sensor Microwave Imagers (SSMI), and the F17 Special Sensor Microwave Imager Sounder (SSMIS).

The F8 operated from 1987 through 1991, the F11 from 1991 through 1995, the F13 from 1995 through 2007, and the F17 from 2008 through 2010 (Cavalieri et al., 1997; 1999; 2011; Parkinson and Cavalieri, 2012). After 2010 and up to 2018, we refer to Parkinson (2019), quoting Cavalieri et al. (1999; 2011).

The data set is generated using the Advanced Microwave Scanning Radiometer - Earth Observing System (AMSR-E) Bootstrap Algorithm with daily varying tie-points. Daily (every other day prior to July 1987) and monthly data are available for both the north and south polar regions. Data are gridded on the SSM/I polar stereographic grid (25 x 25 km) and provided in two-byte integer format.

The sampling from 10/26/1978 to 12/02/1987 is not as continuous as afterwards; there is only one datum every two days and a gap from 12/02/87 to 01/13/1988. We have made our computations with and without that extra decade of data and the results are not significantly altered. Therefore, we select to work on the full data set, starting in 1978 and ending in May 2021 (found on the site https://nsidc.org/data/G02135/versions/3). The data are shown in Figure 1 (NHSI in blue, SHSI in red).

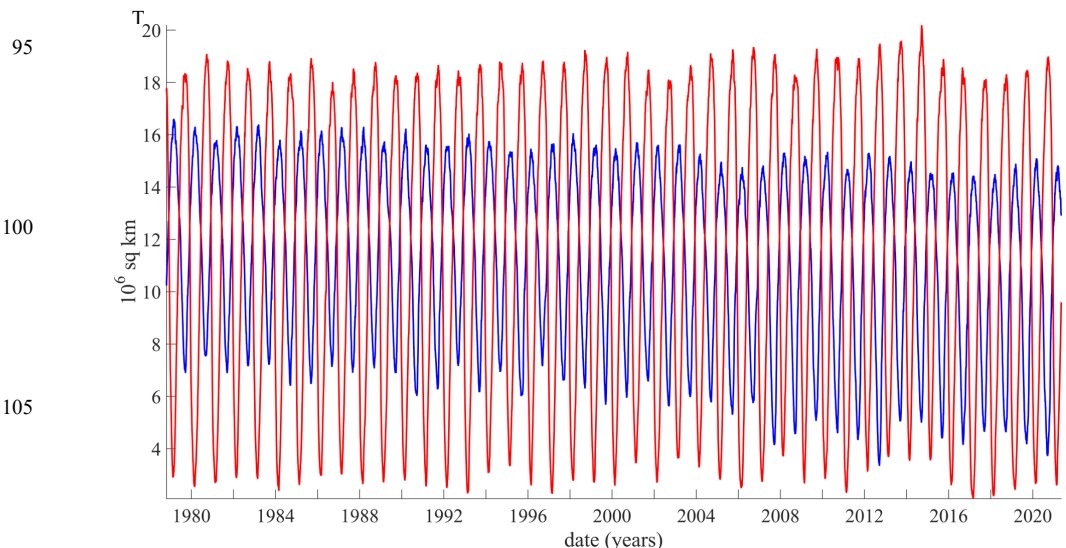

**Figure 1: Raw southern hemisphere (SHSI, red) and northern hemisphere (NHSI, blue) sea-ice data (site https://nsidc.org/data/G02135/versions/3).**

The *pressure data* are maintained by the *Met Office Hadley Centre*[1] and can be accessed in the format of world pressure maps, given every month since 1850. Spatial sampling is on a 5x5 degree grid. The data set is labelled as HadSLP2r (SLP standing for *Sea Level Pressure)*. As mentioned on the *Met Office* site, the series is not homogeneous in time; mean values are homogeneous between 1850 and 2004 and between 2005 and 2020, but variances differ. Allan and Ansell (2004) start with ground observations with location and number shown in their Figure 2. These data undergo a first quality test and then are corrected for a number of local effects, then are homogenized using as a filter *Empirical Mode Decomposition*. We found it very difficult to check all aspects of this complex procedure and prefer, as in most of our previous studies of similar series, to trust the available data. We show in Figure 2 the mean pressures over the same time range as the SI series for latitudes over 60°, SHAP in red and NHAP in blue.

As stated in the introduction, we submit the four time series NHSI, SHSI, NHAP and SHAP to SSA (Golyandina and Zhigljavsky, 2013), using our advanced algorithms as described in Lopes et al. (2017) and Le Mouël et al. (2020b).

---

1    https://www.metoffice.gov.uk/hadobs/hadslp2/data/download.html

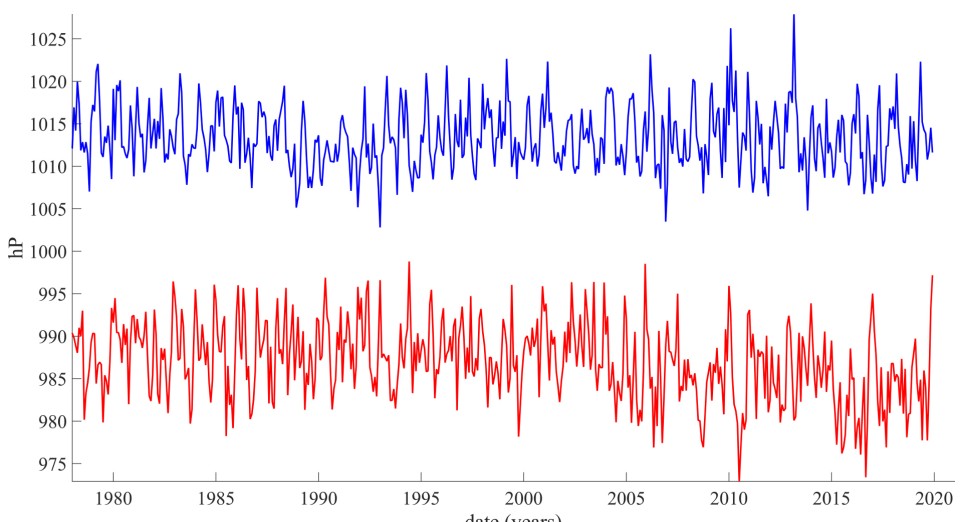

**Figure 2: Mean atmospheric pressures above 60° latitude since 1978: northern hemisphere NHAP in blue, southern hemisphere SHAP in red.**


### 3 Results

The periods (or pseudo-periods) of the main oscillatory components of the series introduced in the previous section are listed in Table 1 (for pressure the components are the same in both hemispheres but for the component with period 0.25 yr marked in red which occurs only in the SH). The uncertainties are estimated based on the half-width of the Fourier

spectral components at half maximum (peak) value. We see in Table 1 that the components with periods of 1 year and their first 4 harmonics (1/2, 1/3, 1/4 and 1/5 year) are common to all series, but for a few exceptions (1/4 yr missing in NHAP not SHAP, also NHSI, 1/5 yr missing in AP, and an extra component a 0.96 yr being present only in NHSI).

For reasons that will appear shortly, we describe and comment in the main body of the paper the components with periods equal of 1 year, 1/2 and 1/3 yr and the trends. Taken together, these four components comprise more than 98% of the

variance of the series. The components with periods larger than 1 year are more briefly discussed in an Appendix.

The *trends* (SSA component 1) are shown in Figure 3, pressure at the top, sea-ice extent at the bottom, from 1980 to 2020. Pressures are given with the mean value subtracted. In each frame the NH is shown in blue, the SH in red. SH pressure decreases in a monotonous way (by 2 hPa or 50 Pa/yr) with a change in slope between 2000 and 2004, whereas NH pressure increases slightly (by 0.2 hPa or 5 Pa/yr). The two intersect in 1990. SI extent increases in the SH (by 15400 km$^2$/yr) and




| | Pressure | | North Hemisphere | | South Hemisphere | |
|---|---|---|---|---|---|---|
| | trend | (99.41 %) | trend | (54.71 %) | trend | (65.48 %) |
| | 44.04 ± 8.35 yr | | | | | |
| | 24.82 ± 2.28 yr | (~0.01 %) | 29.49 yr | (0.03 %) | | |
| | 14.84 ± 0.90 yr | (~0.01 %) | 16.685 ± 5.023 yr | (0.03 %) | | |
| | | | 12.567 ± 2.147 yr | (0.02 %) | | |
| | 4.21 ± 0.90 yr | (~0.01 %) | 5.321 ± 0.397 yr | (0.05 %) | 4.883 ± 0.564 yr | (0.71 %) |
| | | | 2.383 ± 0.082 yr | (0.02 %) | 2.376 ± 0.117 yr | (0.03 %) |
| | 1.84± 0.02 yr | (~0.01 %) | 1.714 ± 0.045 yr | (0.04 %) | 1.749 ± 0.055 yr | (0.03 %) |
| | | | 1.300 ± 0.030 yr | (0.05%) | 1.378 ± 0.034 yr | (0.09 %) |
| | | | | | 1.191 ± 0.073 yr | (0.15 %) |
| | 0.999 ± 0.000 yr | (~0.13 %) | 1.000 ± 0.014 yr | (41.26 %) | 0.999 ± 0.014 yr | (21.73 %) |
| | | | 0.960 ± 0.012 yr | (0.04 %) | | |
| | 0.50 ± 0.000 yr | (~0.01 %) | 0.499 ± 0.003 yr | (1.15 %) | 0.500 ± 0.003 yr | (6.72 %) |
| | 0.33± 0.000 yr | (~0.01 %) | 0.333 ± 0.001 yr | (0.21 %) | 0.333 ± 0.001 yr | (0.52 %) |
| | 0.25± 0.000 yr | (~0.01 %) | | | 0.250 ± 0.000 yr | (0.07 %) |
| | | | 0.200 ± 0.000 yr | (0.21 %) | 0.200 ± 0.000 yr | (0.22 %) |
| | | ~99.70 % | | 97.82 % | | 95.75 % |

**Table 1: Periods (or quasi-periods) of the SSA components of time series (from left to right) AP, NHSI and SHSI in decreasing order of magnitude (with % contribution to the variance). The value listed in red at the bottom of the AP column is a note that this period is encountered only in the SH.**











**Figure 3: Top – Trends extracted by SSA (minus their mean value) of atmospheric pressure for the northern (blue curve NHAP) and southern (red curve SHAP) hemisphere. Bottom – Same for sea-ice extent NHSI and SHSI; same color code. The half sum of both is shown in black.**

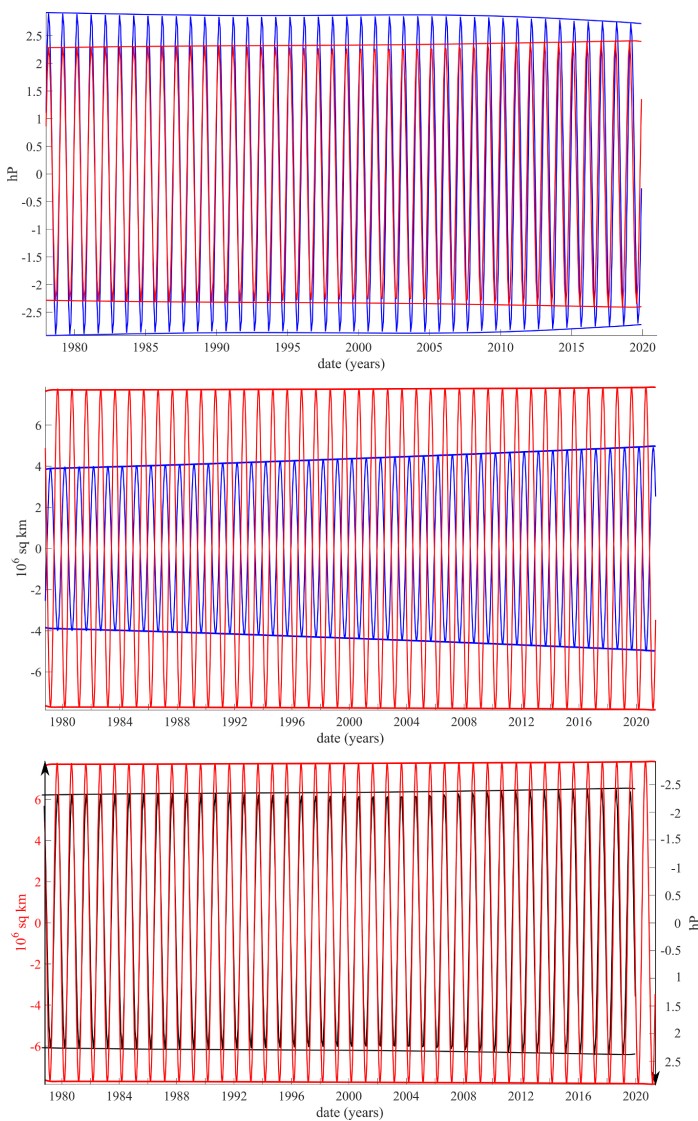

**Figure 4a: Top part - the AP annual component (red for SH, blue for NH); central part - the same for sea ice extent SI; bottom part – comparison of the annual components of SHSI (red) and SHAP (black).**

265

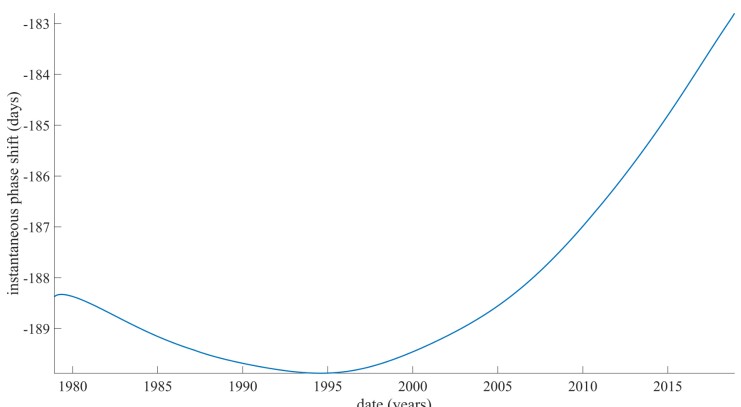

**Figure 4b: Instantaneous phase shift between the annual components of SHSI and SHAP.**

decreases in the NH (by 58300 km$^2$/yr). These values are in good agreement with Parkinson (2019). The two intersect in 1997; SHSI undergoes a slight change in slope around 2013. The two sets of trends in Figure 3 have a degree of anti-symmetry with a phase lag of about 7 years. When hemispheric pressure grows sea-ice extent decreases and vice-versa, but the proportionality factors are very different (12 10$^6$km$^2$/hPa for NH, 0.3 10$^6$km$^2$/hPa for SH). The trend of NHSI represents

68% of NHSI total variance, and that of SHSI 54%.

The next components of the four time series identified with SSA are the *annual components*. They are shown in Figure 4. The top part shows the AP component (red for SH, blue for NH); the central part shows the same for sea ice extent SI and the bottom part compares the annual components for SHSI (red) and SHAP (black). Time runs from 1978 to 2020. The annual components are in perfect phase; the amplitude for NH is constant whereas that for SH decreases slightly over

the 40 years. We have also analysed the full data set of pressure from 1848 to 2020 (not shown here). At the start of this period, NHAP is in advance quadrature with respect to SHAP and slowly drifts, coming to its current "in phase" behaviour seen in Figure 4a. In the case of sea ice, the two annual components are as expected in perfect phase opposition. The NH curve is modulated with a long term (pluri-centennial?) variation. In the bottom part of Figure 4a and in Figure 4b, we see that the annual components of SI and AP are in phase opposition in the southern hemisphere polar regions (the pressure axis

has been reversed in Figure 4a bottom).

Figure 5 (top) shows the close correspondence between the annual component of SHSI (in red) and the variation in Sun-Earth distance (in black). The latter is provided by the institute of Celestial Mechanics and Ephemerid Calculations

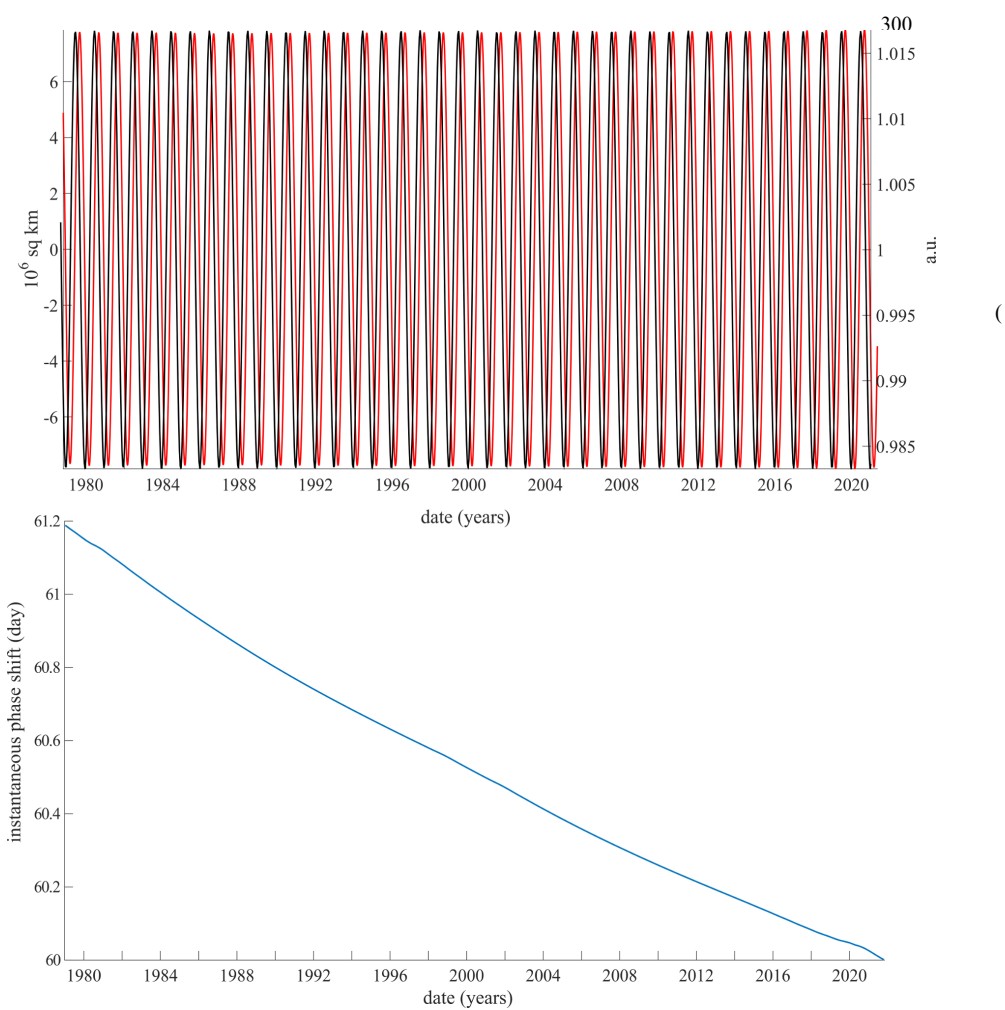

**Figure 5: (top) The variation of Sun-Earth distance since 1978 (in black) is compared to the annual component of SHSI (in red).**
**(bottom) The phase difference between the two.**

330
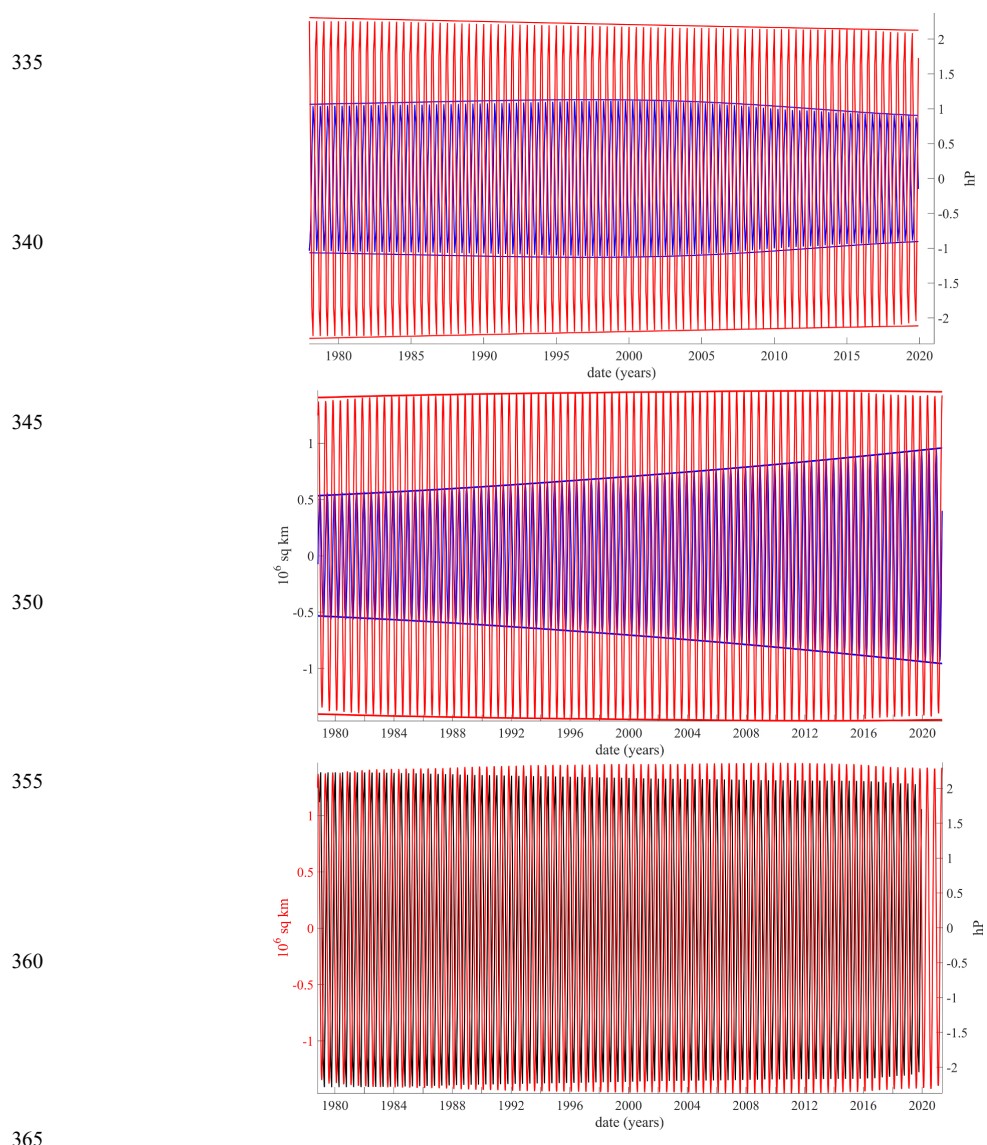

**Figure 6a: Top part - the AP semi-annual component (red for SH, blue for NH); central part - the same for sea ice extent SI;**

**bottom part – comparison of the semi-annual components of SHSI (red)**

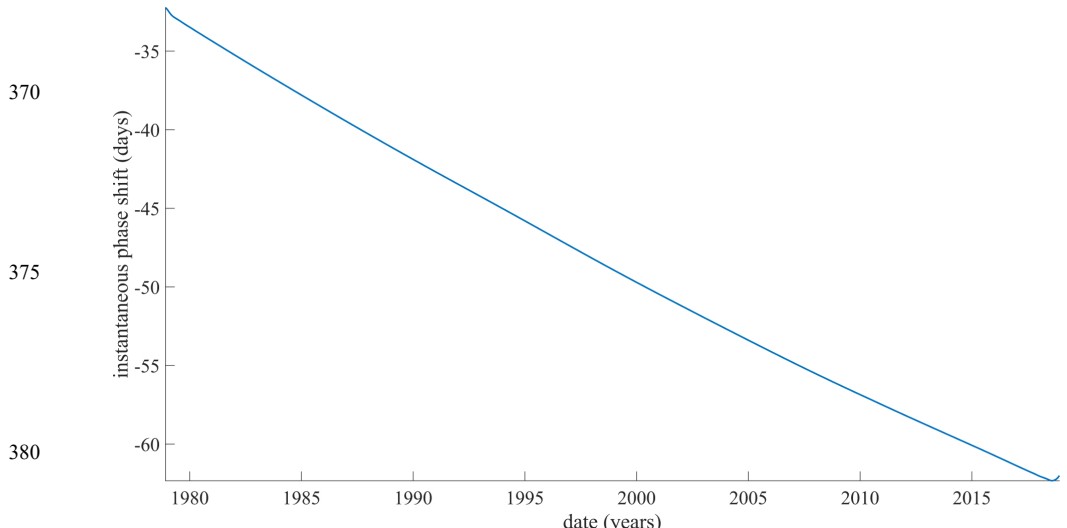

**Figure 6b: Instantaneous phase shift between the semi annual components of SHSI and SHAP**

(IMCCE[2]). This is truly astronomical forcing, in the sense of P.N. Mayaud (pers. comm. 1977: "astronomical" when the forcing is periodical (gravitational) and the spectrum a sharp line, astrophysical when the forcing comes for instance from a turbulent diffusive process (f.i. the solar wind) and the spectrum is broader). The Sun-Earth distance curve always precedes sea-ice formation (and removal) by a constant 61 days (or about 60° since 1 day in a year is about 1°; Figure 5 bottom).

The next oscillating component is the semi-annual cycle shown in Figure 6a. The phase lag between sea-ice extent and

390 pressure decreases from -35 to -60 days (~ degrees; Figure 6b). The NH curves are strongly modulated on durations in excess of a century. The next component (1/3 yr, Figure 7a) is present in sea-ice in both hemispheres, but for pressure it is present in the SH but absent in the NH. The SH AP and SI are significantly modulated. The zoom at the bottom of Figure 7a shows the phase lag, which is the same for the four harmonics (Figure 7b).

The next two components are at 1/4 and 1/5 yr (Table 1). They are shown in Figures 8a and 9a.

In summary, Figures 4 to 9 exhibit a suite of four harmonics of the annual oscillation, in both cases (mean atmospheric pressure and sea-ice extent in both polar regions NH and SH) forced (directly or indirectly) by the annual revolution of Earth about the Sun. A few breaks from symmetry between hemispheres are observed (for 1/3 yr in the case of SI, for 1/4 yr in the case of AP), possibly due to the geography of the continent/ocean outlines in the two polar regions. Also, the pressure harmonics have a saw-tooth pattern of rise and fall rather than a sinusoidal behaviour. At the time scales we are

interested in and with a monthly sampling rate, the atmosphere responds instantaneously to the revolution about the Sun,

2   http://vo.imcce.fr/webservices/miriade/?forms





whence the saw-tooth pattern. In the case of the hydrosphere, notably the ice cover, the response is proportional to the integral of pressure, hence the sinusoidal pattern. The first SSA components (the trends) contain more than half and up to 99% of the total variance of the time series (~68% for NHSI, ~ 54% for SHSI, ~99% for NHAP and ~98% for SHAP**).** The percentages of variances of the fundamental (1 yr) and 4 first harmonics are given in Table 2.

405

|  | SHSI | NHSI | SHAP | NHAP |
|---|---|---|---|---|
| 1 yr | 71.8% | 93.8% | 45% | 45% |
| 1/2 yr | 22.2% | 2.7% | 3.5% | 1.7% |
| 1/3 yr | 1.7% | 0.8% | 3.5% | n.a. |
| 1/4 yr | 0.2% | n.a. | 3.5% | 3% |
| 1/5 yr | 0.8% | 0.8% | n.a. | n.a. |

**Table 2: Percent variance of annual SSA components and their first four harmonics. Columns: the four time series; lines: the five periods.**
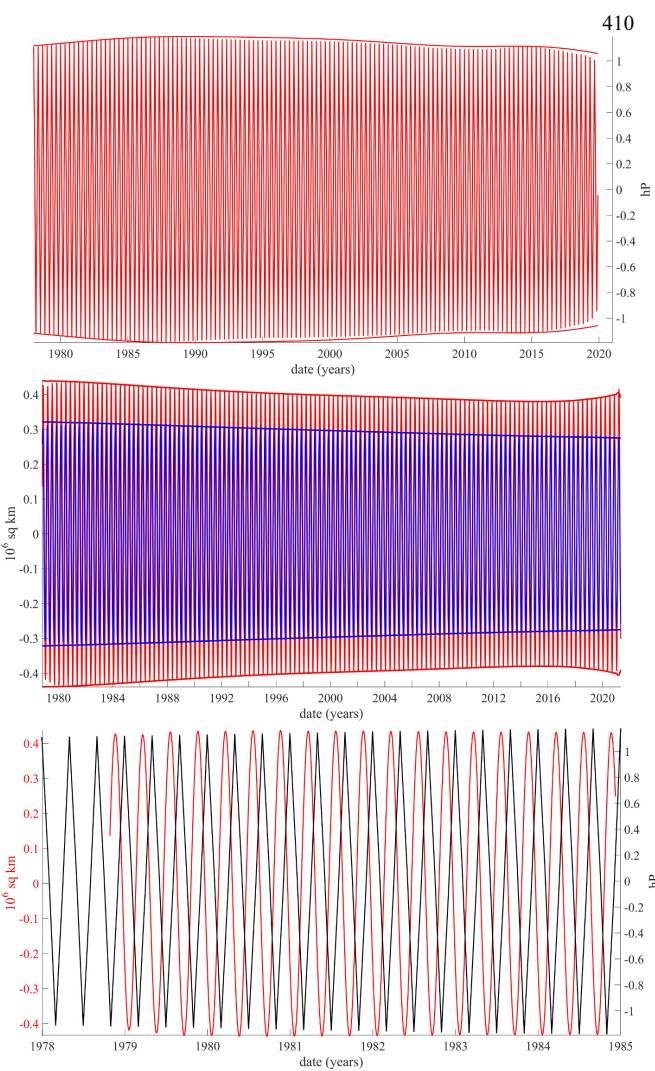

**Figure 7a: Top part - the AP 1/3 yr component for SH (red; it is absent from NH); central part - the SI 1/3 yr component (red for SH, blue for NH); bottom part – comparison of the 1/3 yr components of SHSI (red) and SHAP (black), zooming on the period 1978-1985.**




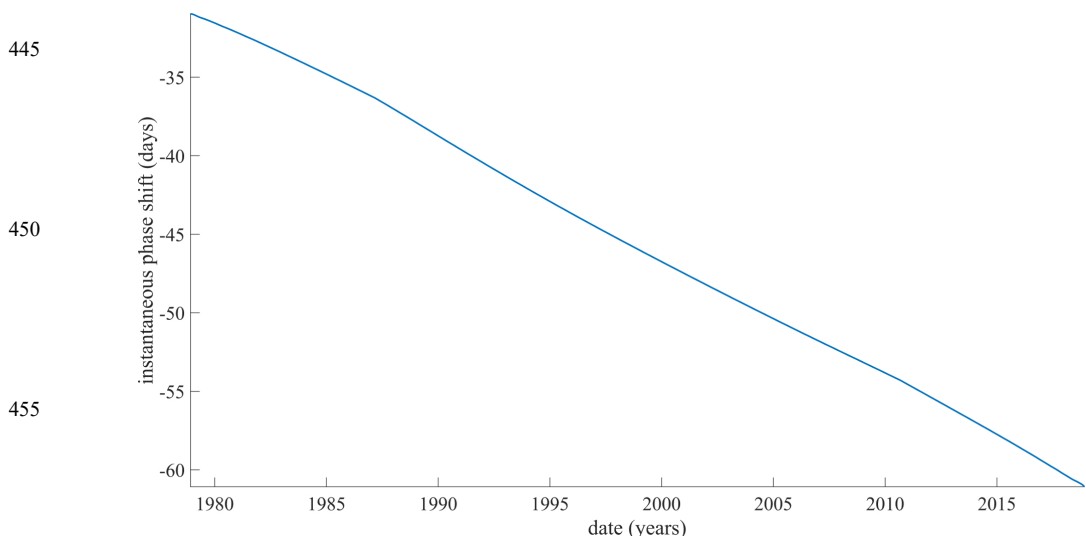

**Figure 7b: Instantaneous phase shift between the 1/3 yr components of SHSI and SHAP.**









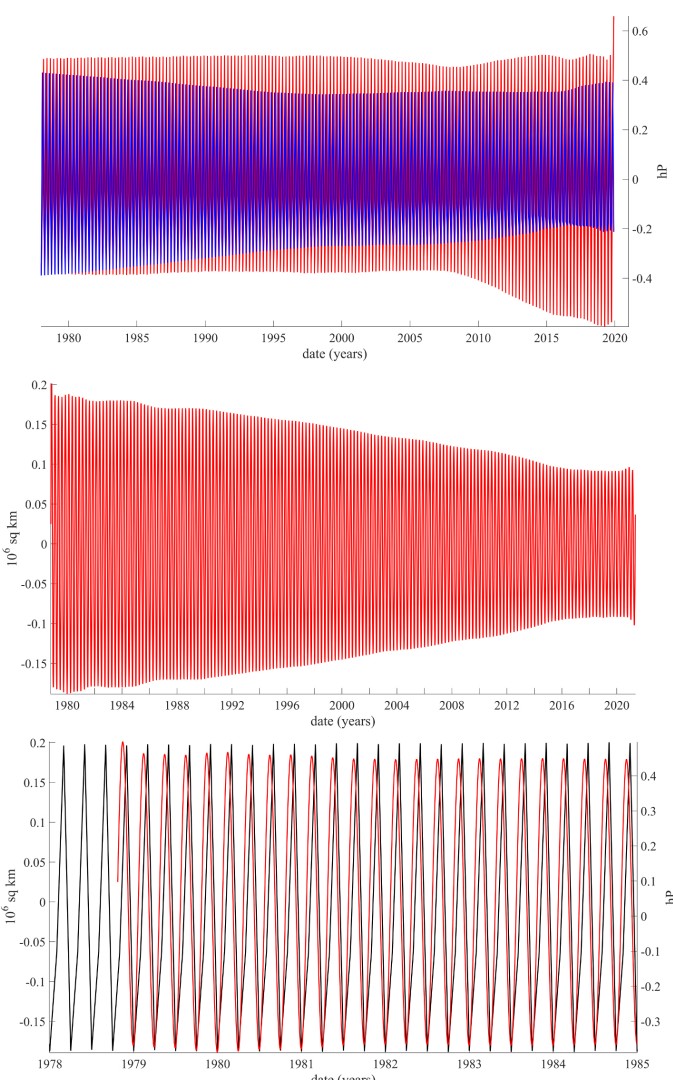

**Figure 8a: Top part - the AP 1/4 yr component for SH (red) and NH (blue); central part - the SI 1/4 yr component for SH (red; it is**

**absent from NH); bottom part – comparison of the 1/4 yr components of SHSI (red) and SHAP (black), zooming on the period**

**1978-1985.**






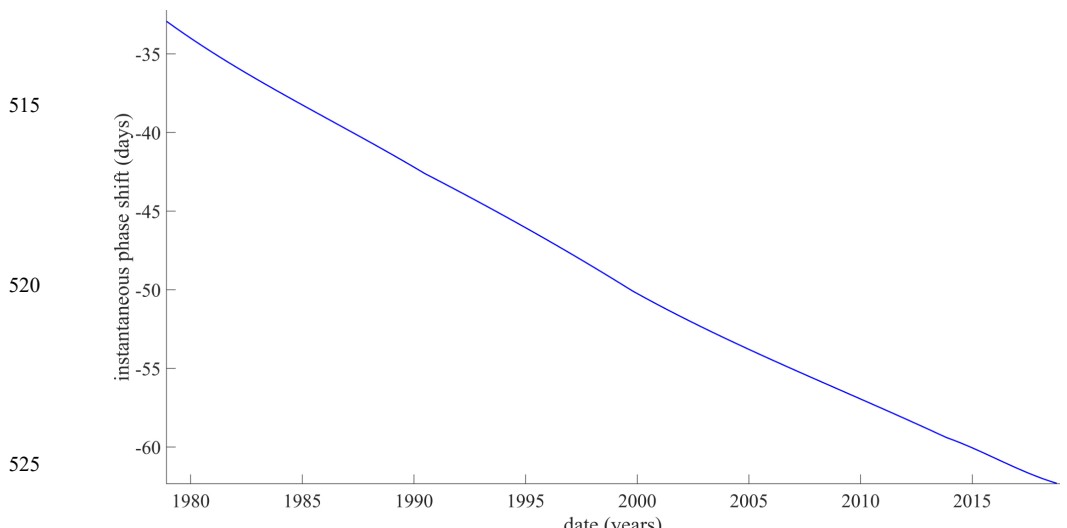

**Figure 8b: Instantaneous phase shift between the 1/4 yr components of SHSI and SHAP.**





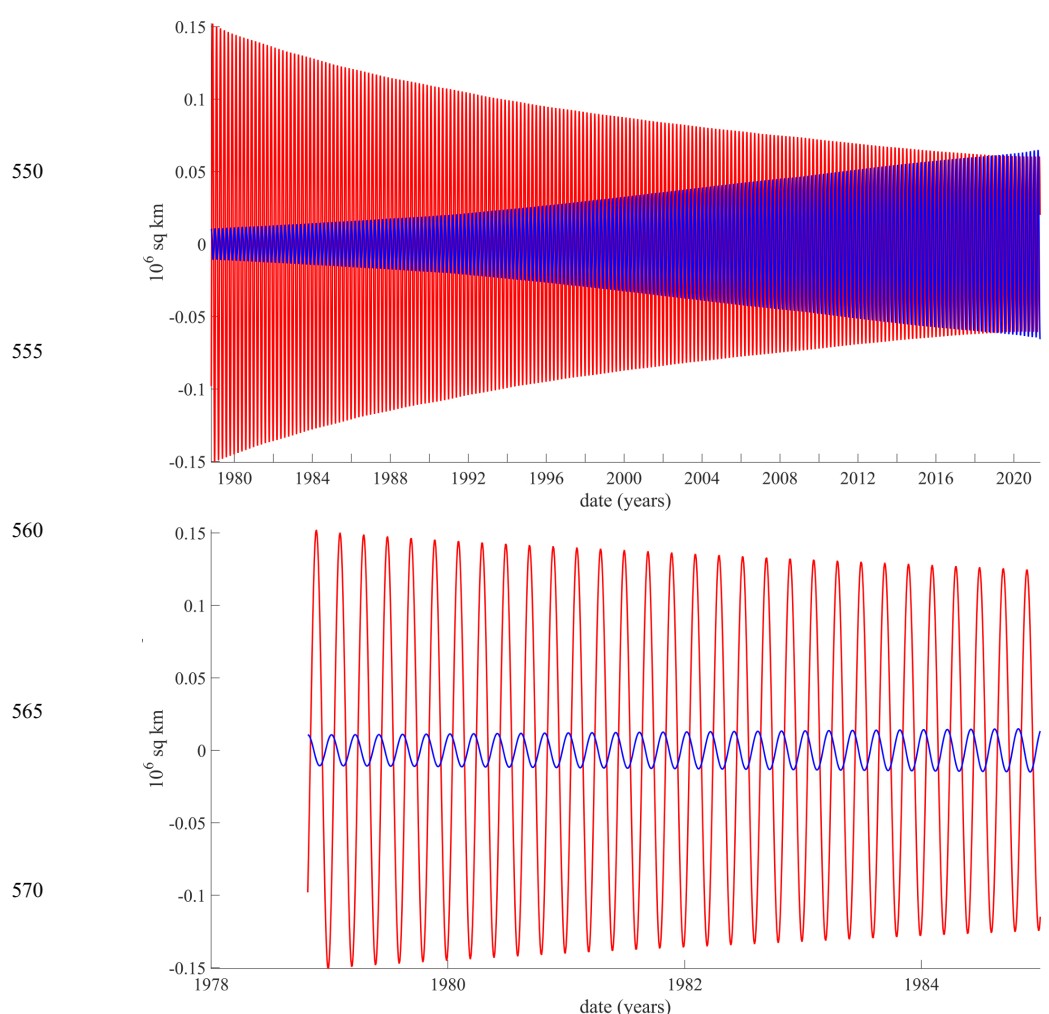






**Figure 9a: Top part - the SI 1/5 yr component for SH (red) and NH (blue); bottom part – same, zooming on the period 1978-1985.**


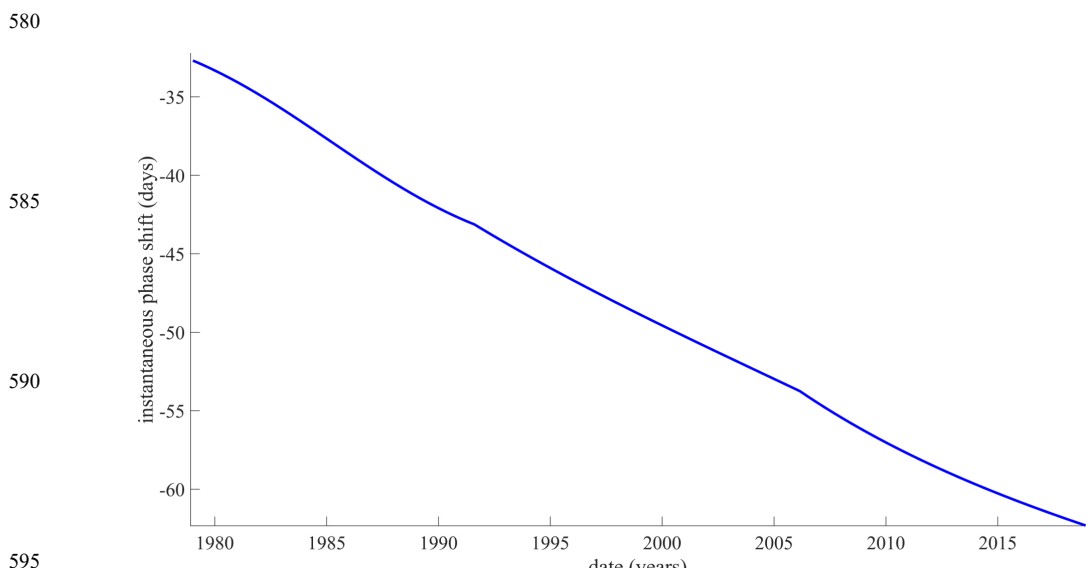

**Figure 9b: Instantaneous phase shift between the 1/5 yr components of SHSI and SHAP.**

**4 The Case of Antarctica**

It is now well known that, whereas the Arctic region is subjected to melting, the case of the Antarctic is more complex, with regions of dominant melting and regions of increase in sea-ice extent, the overall SI budget being an increase. A number of authors have divided the Antarctic in zones (*e.g.* Zhang, 2007; Holland, 2014; Turner et al., 2014, 2015, 2020; and Parkinson, 2019, from which Figure 10a comes). We have divided the pressure data set as a function of season; we use the full 170 years (since 1846) over which AP is available. We have seen in the previous sections that together the trend and

annual oscillation contain more than 98% of the original pressure series variance.

The maps for the northern hemisphere are very similar at all four seasons. They are so similar and stable over the 170 years of data that they are almost identical to their mean. The resulting map for the southern hemisphere is shown in Figure 10b. It features three major high-pressure maxima over the South Atlantic, South Indian Ocean and South East Pacific. A noticeable feature of the Southern Hemisphere maps is a strong pressure minimum anomaly offset from the South Pole and

located between the Ross and Weddell Seas, encompassing the Antarctic peninsula and Trans-Antarctic Mountains.

The strong pressure anomaly straddling the Antarctic peninsula separates four zones of increase in sea-ice extent (Weddell Sea, Indian Ocean, W Pacific Ocean and Ross Sea) from a zone of melting over the Bellingshausen and Amundsen Seas (Figure 10a, from Parkinson, 2019). The amplitude of the trend anomaly is 20 hPa and that of the annual component 10




hPa. This pressure anomaly, the largest in the SH South of 50°S latitude, is connected to the topography of the West
Antarctic Peninsula. It corresponds to the Ross Sea "dipole" of Bertler et al. (2018). We retain the term "dipole" of Bertler et
al. (2018), although this anomaly doesn't have the properties of a physical dipole.

Bertler et al (2018) document the persistent presence of the Ross "dipole" anomaly over the last 2700 years. In the
past three centuries, the "dipole" has been evidenced on the basis of coherency between sea ice extent and temperature
(estimated from isotopic measurements δD) (Figure 10c). The sea ice extent increases in the Ross Sea and decreases in the
Bellingshausen Sea. The same "dipole" feature (anomaly) is present in the pressure map (Figure 10b, this paper) and the
winds (Figure 10d, Holland et al., 2017).

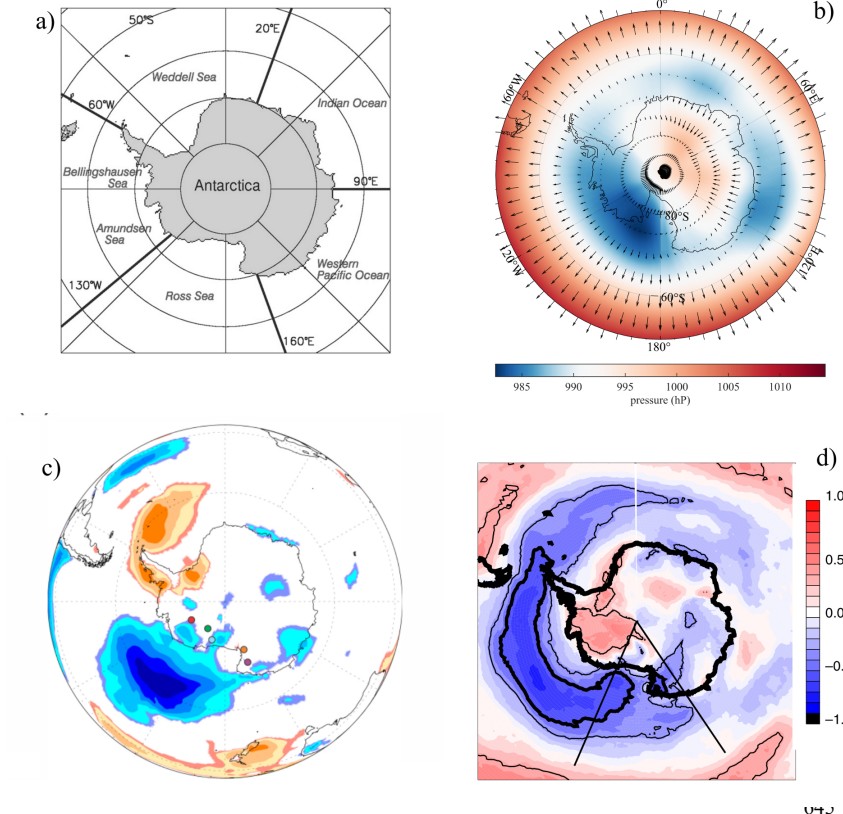




**Figure 10: a) Antarctica divided in five sectors, from Parkinson, 2019. b) Mean of SHAP since 1750. c) Correlation between temperature in the past 2700 years (based on δD) and current sea ice extent, from Bertler et al., 2018. d) Correlation between winds and Sea Ice 1979-2015, from Holland et al., 2017.**

**5 Discussion and Concluding Remarks**

In this paper, we have analysed in parallel, using Singular Spectral Analysis, variations of (1) the Arctic and Antarctic sea-ice extents, and (2) the atmospheric surface pressure in both hemispheres.

**5.1 Sea-Ice.** The ice-extents are dominated by a quasi-linear trend over the 42 yr period when data are available
(1978-2020) and an annual component. Taken together, these two components represent more than 98% of the signal variance (Table 1). A key observation is that both ice-extent series share the same set of harmonics of the annual component (1/2, 1/3, ¼ and 1/5 yr), although they are much less intense than both the trend and annual component.

The multi-decadal trends of sea-ice extent in the Arctic and Antarctic are of opposite sign. Therefore, it cannot be concluded without caution that atmospheric and oceanic warming are the sole cause.

The periods extracted by SSA analysis are all either "astrophysical" or "astronomical" in the sense of Mayaud (see above). The remarkable series of harmonics of 1 year are consequences of the Earth's revolution about the Sun. They are therefore astronomical. The components with period longer than a year (see Table 1 and Appendix) form a possible set of even harmonics of the Schwabe cycle and argue for a source in the Sun or in Jupiter (Courtillot et al., 2021; Lopes et al., 2021). They are therefore astrophysical. We have seen in a previous study (Le Mouël et al., 2020b) that the SSA components
of global surface temperature have mainly "solar" characteristics (~22, 11, 9, 5.5, 4.7 yr).

We propose a hypothesis concerning the often quasi linear (and in all cases low-degree) trends. Since they are defined over only 42 years, they could well be incomplete segments of components with periods much longer that the data interval (e.g. the Gleissberg or Hallstadt cycles). The Hallstatt cycle has long been known in the Arctic (*e.g.* Campbell et al., 1998; Darby et al., 2012), but has yet to be evidenced in the Antarctic.


**5.2 Atmospheric pressure.** The pressure series also exhibits the same harmonics of 1 year (except the last and smallest one). This remarkable set of harmonics is not found in the many series we have previously analysed (Courtillot et al., 2013, 2021; Lopes et al., 2017, 2021; Le Mouël et al., 2019a, 2019b, 2019c, 2020a, 2020b, 2021). This argues in favour of a special connection between variations in *pressure* and *sea-ice* extent: either a third physical process forces them both or
one has a forcing effect on the other. Geographical pressure structures are remarkably stable on a decadal to centennial time scale and exhibit symmetries that fluid mechanics could explain. Particularly noteworthy is the three-fold symmetry of pressure trends. In the case of the northern hemisphere, there is a strong order-3 symmetry pinned by the North Atlantic, North East Pacific and central Asia (Tibet). That geometry is constrained by the coastlines and topography. In the southern





hemisphere that order-3 symmetry is somewhat distorted, with the three maxima pinned by the southern oceans (Atlantic,
Indian and SE Pacific; Figure 10b).

**5.3 The case of Antarctica.** This comes in contrast to the (asymmetric) Ross-Weddell "dipole" pressure anomaly.
Figure 10 shows that this anomaly is seen in correlations of variations in sea-ice extent with atmospheric pressure, surface
temperature (if correctly deduced from oxygen isotope anomalies) and winds. The pressure anomaly over the Antarctic
Peninsula contrasts with the order-3 symmetry but fits topographic forcing.

We find a phase lag of -186 days for the annual component (quasi-phase opposition, Figure 4b), and the same
decrease from -30 to -60 days over 42 years for the four harmonic components of SHSI and SHAP with periods ½, 1/3, ¼
and 1/5 yr (Figures 6b, 7b, 8b, 9b). This could be interpreted as a slowly increasing lag between pressure variations, the
forcing factor linked primarily to the Earth's revolution about the Sun, and the resulting sea-ice extent variations.

Robin (1977) observed a strong correlation between temperature, rainfall and accumulation rate of ice in Antarctica.
The (negative) sign of the trend of pressure and (positive) sign of the trend of the temperature change are opposite to what
might have been expected. The relative importance of warming over the past 50 years (~1°/15°=7%) vs the change in
pressure (~1hPa/1000hPa=0.1%) which is two orders of magnitude smaller (Figure 11). This relatively strong warming
should have a larger effect on pressure.


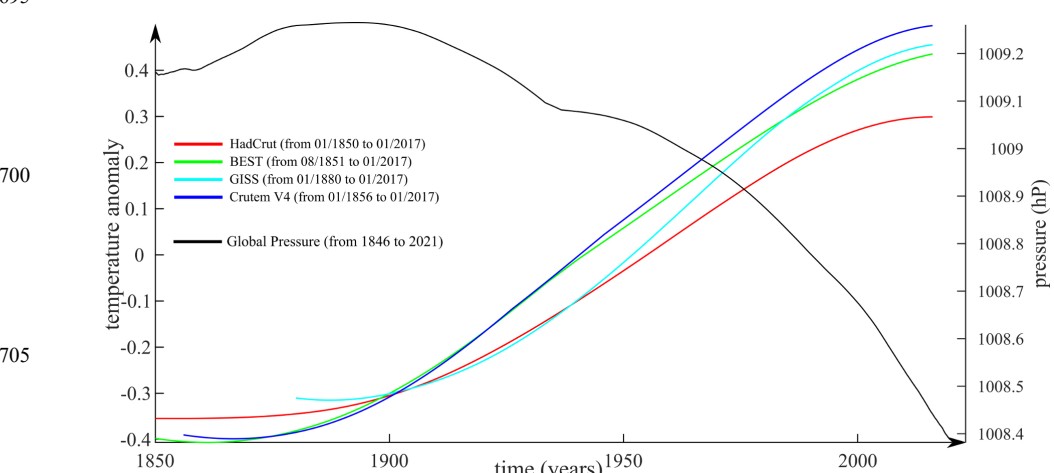

**Figure 11: SSA trend of four models of global surface temperature change (Le Mouël et al., 2020b) and of global atmospheric**
**pressure (this paper) from 1850 to 2020.**



**5.4 Which mechanism?** In both hemispheres the sign of the trend of sea-ice extent is *opposite* to that of pressure (Figure 3). In terms of intensity, the relative change in pressure over the past 40 years is on the order of +0.3 hPa (0.03%) for the NH and -2 hPa (-0.2%) for the SH, when that of sea-ice extent is -2.4 $10^6$ km$^2$ (-12%) for the NH and +0.7 $10^6$ km$^2$ (3.5%) for the SH. The ratio of relative changes of sea-ice extent vs pressure is 400 for the NH and 17 for the SH. This certainly does not argue for a simple mechanism relating atmospheric pressure change to variations in sea-ice cover. Yet we have seen the similar presence of the annual component and its first four harmonics in both time series. We hope that the remarkable sets of SSA components can be used in understanding the mechanisms that govern changes in sea-ice extent. We close on some reflexions that might indicate a path to a solution.

When attempting to study the motions of fluid masses on Earth (core, mantle, hydrosphere, atmosphere) one may resort to the paradigm of stationary turbulent flows (*e.g.* Chandrasekhar, 1961; Fisch, 1995). This is also the case for solar physics (*e.g.* Charbonneau, 2014; Le Mouël et al., 2020). The Taylor-Couette flow is considered for the large atmospheric cells, Hadley, Ferrel and Polar at increasingly high latitudes. Analytic solutions are well known in the case of cylindrical coordinates (*e.g.* Taylor, 1923), but raise major problems in the case of spherical geometry (*e.g.* Schrauf, 1986; Mamun and Tuckerman, 1995; Nakabayashi and Tsuchida, 1995; Hollerbach and Egber, 2006; Malhoul et al., 2016; Garcia et al., 2019; Mannix and Mestel, 2021). One must solve the most general form of the Navier-Stokes equation (Taylor, 1923; Landau and Lifchitz, 1959, chapter 27 ; Chandrasekhar, 1961, chapter 8).

In the case of a cylindrical geometry, the solution has the form:

$$\mathbf{v}(r,\phi,z)=exp\ i(n\phi+kz-\omega t)\mathbf{f}(r) \qquad (1)$$

where **v** is the velocity of the perturbation component of the flow, and **f(r)** the unit vector of its direction. Wavenumber $k$ is associated with the coordinate $z$ on the axis of the cylinder and takes continuous values, $n$ is an integer (the order-3 symmetry we find in Figure 10 would be a case when $n$=2). When $n$ and $k$ are fixed, $\omega$ is in the form of a discrete series of eigen-components $\omega_j(k)$. The solution oscillates with boundary conditions on the outer and inner cylinders and $div\mathbf{v}$=0.

An analytic solution of **f**($r$) can be found for a cylinder, but is very complex in the case of a sphere. We can only attempt qualitative suggestions. $\omega$ is constrained by the rotation of Earth and its revolution about the Sun to first order. Hence, it is not surprising to find the harmonic series of eigen frequencies with periods 1, ½, 1/3, ¼ and 1/5 year (there is no reason for it to stop there but the following terms may be too small to detect with our data). We recall that to first order polar motion comprises a secular trend (the Markowicz drift), a forced annual oscillation and the Chandler wobble (e.g. Lambeck, 2005 ; Lopes et al., 2017).

A recent numerical statistical study of flow in an entirely fluid sphere, using spherical harmonics, shows flow anomalies at high latitudes that evolve into bands as rotation velocity is increased (Supekar et al. 2020). "Dipole" (anomalous) flow structures appear near the poles and may persist as velocity is increased. They look somewhat like the pressure polar anomalies that are seen in the Antarctic.




**Appendix: SSA components with periods longer than 1 year**

These components are listed in decreasing order of period in Table 1. Because the SI data are available only since 1978, periods longer than ~42/2=21 years can only be regarded with caution. The total variance encompassed in these components is 0.04% for pressure, 0.24% for NHSI and 1.11% for SHSI. The component of AP at 14.8±0.9 yr can be associated with that

of SHSI at 12.6±2.1 yr, and could correspond to the Schwabe ~11 yr cycle. Then there is a series of four successive components that could correspond to even harmonics of the Schwabe cycle: 11/2, 11/4, 11/6 and 11/8 yr (in that order the pairs for NH/SH are 5.3±0.4/4.9±0.6; 2.4±0.1/2.4±0.1; 1.7±0.05/1.7±0.05; 1.3±0.03/1.4±0.03). These are not seen in pressure but are common to the two polar regions. They may indicate a very small but consistent solar (or Jupiter) effect. One could also argue that the series of longer periods 24.8±2.3 yr and 14.8±0.9 yr in AP are a trace of the Schwabe and Hale

solar cycles.

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
