# Peer review of "A strong link between variations in sea-ice extent and global atmospheric pressure ?"

_The Cryosphere, 2021_

## Referee Comment (RC1)

Authors used Singular Spectral Analysis to examine variations of the Arctic and Antarctic sea-ice extents (SI), and of the atmospheric surface pressure (AP) in both hemispheres (NH and SH). There exists a range of periods (an annual, 1/2, 1/3, 1/4 and 1/5). Some relationships between sea ice and pressure are built. The manuscript was written well. Some improvements need to be made before the manuscript is accepted. The results presented in this study are statistical, the mechanisms behind the results lack. This is the main shortcoming.

For SSA method, what is the difference between SSA and other methods (Circulant SSA, EMD)?

Lines 38-42 Authors should gave some reasons for the dissymmetry of Arctic and Antarctic sea ice extent.

Lines 47-54 The large-scale atmospheric oscillation in the Arctic is not only AO. The factors influencing the Antarctic climate include forcings in the Indian, Atlantic, and Pacific Oceans. Authors only introduced Pacific factors.

While authors represent the results of the SSA analysis for a given period ( for example annual cycle), please display the spatial pattern of pressure and sea ice at maximum and minimum values, and explain the relationship between air pressure and sea ice pattern.

Authors exhibited the 1/2 year period. There is a common phenomenon in the Southern Ocean. There are a lot of literatures.

"The semi-annual oscillation (SAO) in the middle and high latitudes is an important and well known component of the Southern Hemisphere climate. An overview of the early literature on the SAO is given by van Loon (1967), and a reexamination of the phenomenon and its causes is presented by Meehl (1991). "

The semi-annual oscillation and Antarctic climate Part 1-4 depict SAO and its effect on the Antarctic climate.

VA N LOONH, . 1967. The half-yearly oscillation in the middle and high southern latitudes and the coreless winter. Journal of Atmospheric Sciences. 24, 472-486.

MEEHLG, .A. 1991. A reexamination of the mechanism of the semiannual cyclc in the Southern Hemisphere. Journal of Climate, 4, 91 1-926.

For 1/3, 1/4, and 1/5 period, what mechanisms behind these periods are there?

Lines 389-390 "The phase lag between sea-ice extent and pressure decreases from -35 to -60 days (~ degrees; Figure 6b)." Pressure precedes sea ice extent 35-60 days for semi-annual period? Why? There is an increasing trend. Why?

The similar case also occurs for 1/3, 1/4 and 1/5 annual periods in Figure 7b, 8b and 9b.

There is no color bar for Figure 10c.

---

## Community Comment (CC4)

Reply to Comment tc-2021-216-RC1

Q1: Authors used Singular Spectral Analysis to examine variations of the Arctic and Antarctic sea-ice extents (SI), and of the atmospheric surface pressure (AP) in both hemispheres (NH and SH). There exists a range of periods (an annual, 1/2, 1/3, 1/4 and 1/5). Some relationships between sea ice and pressure are built. The manuscript was written well. Some improvements need to be made before the manuscript is accepted. The results presented in this study are statistical, the mechanisms behind the results lack. This is the main shortcoming.

A1: We can only agree. So far, we have not yet been able to produce a full model based on a precise mechanism. We do no perform a «statistical analysis». We extract in an objective way (SSA) periodical or quasi-periodical cycles from two a priori distinct data sets. Our «observations only» conclusion is the remarkable similarity (not to say identity) of the two sets of periods, which implies that some explanation must be found. We propose hypotheses but the observations are important enough in our view to be shared, so that others have a chance to uncover the full mechanism.

Q2: For SSA method, what is the difference between SSA and other methods (Circulant SSA, EMD)?

A2: There are roughly three families of tools that can be used to analyze a signal (time series): Fourier analysis, wavelets and what we will call «*ad hoc*» analyses. For the first two, the signal is projected on a basis of imposed orthogonal functions. In the ad hoc analyses, the orthogonal basis is built from the information carried by the signal itself. These often take advantage of the properties of embedding, as exhibited by the descending Toeplitz or Hankel diagonal matrices (*e.g.* Lemmerling and Van Huffel, 2001) combined with the powerful Singular Value Decomposition (*e.g.* Golub and Reinsch, 1971).

These three approaches should not be considered as opposed: given sufficient patience one

can obtain the same results in the end. The reason why Fourier invents the method that bears his name is because he has to solve a precise set of physical/mathematical problems: uncoupling the variables in the heat equation, and expressing the solution as a combination of pure oscillations, namely the forcing of daily variations (*e.g.* Fourier, 1822). The Fourier transform can be seen as a kind of cross correlation of the signal with an infinite series of different frequencies. This requires that for its all duration the signal be stationary. This property is clearly not that of seismic waves propagating in a heterogeneous, attenuating and dispersing ground. This is why Jean Morlet modified the infinite sinuses into «Morlet» wavelets (*e.g.* Morlet et al., 1982). The two methods appear for different problems in different physical cases and are pushed to their limits. Geophysicists (again) introduced a new method, Singular Spectrum analysis (SSA), in order to fill gaps in paleoclimatic sedimentary series, when none of its statistics is known (*e.g.* Vautard and Ghil, 1989). As briefly mentioned above, in these ad hoc methods the basis on which the signal is projected is built from the information contained in the signal. Therefore (unlike in Fourier analysis) one cannot predict whether a certain eigenvector will have a given pure frequency, and unlike in wavelets one cannot know ahead of time whether a certain amplitude characteristic will be in such or such wavelet scale.

What is the physical reality embedded in each of these transformations (what is the physical nature of a given component)? The answer is simple: none. The only goal is to distribute the energy contained in a signal over a new reading grid that may better allow to answer the original questions. As another example, cross correlating the horizontal component of the geomagnetic field at a given observatory with an infinite sinus («beginning» at minus infinity and «ending» at plus infinity) has no physical reality. Then, as we do not impose a perfect orthogonal basis, how can we be sure that our eigenvectors (that are orthogonal having been obtained with SVD) are the most «perfect» as possible? This is the question of separability in SSA (treated for example by Golyandina and Zhigljavsky, 2013; chapters 2.3.3 and 2.5.4).

Without going into too much detail, another way to treat this problem is to ask within the

frame of SSA how can our Hankel/Toeplitz matrices be the best possible. There are several approaches. We have selected «iterative SSA» that, through iterations of rotations in the space of eigenvectors associated with (f.i.) a «varimax» criterion, optimizes the separability of components (e.g. Hubert, 1985).  In conclusion, given its results, SSA is a remarkably powerful method in certain specific cases but it is in no way a «express button» tool.

For this reason (lack of «automatizability»), Bógalo et al (2021) recently proposed the new «circulating» SSA (cSSA). When classical SSA builds its Hankel/Toeplitz matrix from signal values (lines and columns are elements of the signal), cSSA builds a matrix in which lines and columns are auto-covariances (**E**) of the signal:

$$\gamma_m = \int_0^1 f(\omega) exp(i * 2 * \pi * m * \omega) = E\{x_t, x_{t-m}\}$$

These are used to force the automatic extraction of the most oscillating components. The new estimator, the covariance, carries with it the whole problems linked to estimators in general (cf. Claerbout, 1976; Papoulis, 1984).

Again, these methods should not be opposed one to the other : iSSA has been developed for geophysical problems, cSSA for economic questions. If the various « recipes » are followed carefully, extractions of components should end in the same way.

Let us end with Empirical Mode Decomposition (**EMD**). Here again, a geophysicist investigates non stationary signals (Huang et al., 1998). The only point that is common with SSA is that orthogonal bases are directly built from the signal, but are the most oscillating as possible, contrary to SSA, using the properties of the Hilbert transform. These are used to calculate the envelopes and instantaneous phases of any signal. Both EMD and SSA are less sensitive to stationary signals, unlike Fourier. Huang et al., (1998) calculate intrinsic mode functions (IMF, whose properties they define) along the time space signal, leading to the following algorithm : One first determines the maxima ogf the upper envelope, interpolates them with splines and the same is done for the lower envelope. The two curves must contain the whole of the signal. At the first step, the difference between the two envelopes and the original signal is the first mode (or component). Then one

iterates. Their may be problems with the regularity of the Hilbert transforms and the spline interpolations (Claerbout 1976) inducing errors in the component amplitudes. Once again, is applied correctly, EMD and SSA should give the same answer.

This short and simplified «tutorial» hopefully answers the referee's central question. The only real differences lie in the scientific questions that are being asked and led to their construction.

Q3: Lines 38-42 Authors should gave some reasons for the dissymmetry of Arctic and Antarctic sea ice extent

A3: This is not a simple question to answer. Several authors (cited in the paper) have attempted to answer in a semi-qualitative way. Reasons for the dissymmetry include ocean/continent boundaries (grounding ice vs sea-ice). The resulting wind and current patterns are also important. The quasi-circular shape of the Antarctic continent, ice extent, free sea-passage contrasts with the lack of an easy sea-passage in the Arctic. Does the referee suggest adding a few lines along this?

Q4: Lines 47-54 The large-scale atmospheric oscillation in the Arctic is not only AO. The factors influencing the Antarctic climate include forcings in the Indian, Atlantic, and Pacific Oceans. Authors only introduced Pacific factors.

A4: This is correct. But in Le Mouël et al. (2019) we showed that most if not all climatic indices reveal the same cycles (periodicities). We can add a few lines and repeat this reference.

Q5: While authors represent the results of the SSA analysis for a given period ( for example annual cycle), please display the spatial pattern of pressure and sea ice at maximum and minimum

values, and explain the relationship between air pressure and sea ice pattern

A5: Only the pressure patterns are available in 2D. Sea-ice extent is a 1D series that does not allow the interesting exercise suggested by the reviewer. And so far we acknowledge that we do not have a clear physical mechanism to explain the relationship between air pressure and sea ice pattern. We hope some of the readers of our paper (observations) will be inspired bu them to hypothesize a mechanism…

Q6: Authors exhibited the 1/2 year period. There is a common phenomenon in the Southern Ocean. There are a lot of literatures

A6: Indeed. The ½ year period is for instance also found in polar motion (length of day) (Le Mouël et al, 2019b), in sunspots (Le Mouël et al., 2020; Courtillot et al., 2021), in the magnetic field (Cliver  et al., 2004; Le Mouël et al., 2019c) and of course in the Sun-Earth distance.

Q7: "The semi-annual oscillation (SAO) in the middle and high latitudes is an important and well known component of the Southern Hemisphere climate. An overview of the early literature on the SAO is given by van Loon (1967), and a reexamination of the phenomenon and its causes is presented by Meehl (1991)".  The semi-annual oscillation and Antarctic climate Part 1-4 depict SAO and its effect on the Antarctic climate.

A7: We agree with this remark pointed out by the reviewer. The 6-month component plays an important role in climatology, being present for instance in climate indices as well as pressure variations. Actually, this oscillation has been encountered in many other fields. First of all in geomagnetism: we know since Bartels (1932) that magnetic indices over the entire Earth record this component, whose amplitude varies with latitude. Bartels concluded in favor of an internal origin, strongly linked to the revolution around the Sun. We also find a 6 month periodicity in the length of

day (which is a global parameter, e.g. Lambeck, 1980), in sunspots numbers (e.g. Lockwood, 2001) and of course, even if it is modest, in the Earth-Sun distance variation. In summary, the 6 month component is found in many different geophysical and heliophysical fields. We argue that there may exist a general forcing mechanism with a sequence of harmonic components at 1, 1/2, 1/3, 1/4, 1/5 years. This sequence could help us (or others) to uncover its nature.

Q8: For 1/3, 1/4, and 1/5 period, what mechanisms behind these periods are there?

A8: Indeed an important question To our knowledge only the mechanics of turbulent fluids could explain this observation (in a geophysical context). If a rotating sphere or cylinder is forced with a frequency w, this frequency and its harmonics should be encountered in the movement of the fluid. We briefly address this in paragraph 5.4 page 23. We note that whereas there is a unique solution in the case of the cylinder, the problem for a sphere has still to be solved. This suggestion is certainly not the only one and we hope other suggestions will be encouraged by our results.

Q9: Lines 389-390 "The phase lag between sea-ice extent and pressure decreases from -35 to -60 days (~ degrees; Figure 6b)." Pressure precedes sea ice extent 35-60 days for semi-annual period? Why? There is an increasing trend. Why?

A9: Except for the fundamental 1 yr, all harmonics have the same phase lag of 30 days in 42 years. But this is calculated as a phase difference between two signals using a Hilbert transform. All we can say is that the sign of the phase difference indicates that over the analyzed window pressure variations precede sea-ice extent variations. We would need to know the data series from a 0 time origin. Still, the phase lag of 30 days in 42 years is a robust physical estimate that must have a physical origin/explanation. We now must seek which one. Same general conclusion: we have exciting and robust observations that indicate an as yet unknown physical mechanism, most likely contained in Laplace's celestial mechanics.

REFERENCES

Bartels, J., "Terrestrial-magnetic activity and its relations to solar phenomena", *Terrestrial Magnetism and Atmospheric Electricity*, *37*(1), 1-52, 1932.

Bógalo, J., Poncela, P., & Senra, E., "Circulant Singular Spectrum Analysis: A new automated procedure for signal extraction", *Signal Processing*, *179*, 107824, 2021.

Claerbout, Jon F. *Fundamentals of geophysical data processing*. Vol. 274. McGraw-Hill, New York, 1976.

Cliver, E. W., L. Svalgaard, and A. G. Ling. "Origins of the semiannual variation of geomagnetic activity in 1954 and 1996." In *Annales Geophysicae*, vol. 22, no. 1, pp. 93-100. Copernicus GmbH, 2004.

Courtillot, V., F. Lopes, and J. L. Le Mouël. "On the prediction of solar cycles." *Solar Physics* 296.1 (2021): 1-23, 2021

Fourier, J., "Theorie analytique de la chaleur", par M. Fourier. Chez Firmin Didot, père et fils. 1822.

Golub, G. H. et C. Reinsch, "Singular value decomposition and least squares solutions", in Linear Algebra, pp. 134–151. Springer, 1971.

Golyandina, N and Zhigljavsky, A., "*Singular spectrum analysis for time series*". Springer Berlin Heidelberg, 2013.

Huang, N.E., Shen, Z., Long, S.R., Wu, M.C., Shih, H.H., Zheng, Q., Yen, N.C., Tung, C.C. and Liu, H.H., "The empirical mode decomposition and the Hilbert spectrum for nonlinear and non-stationary time series analysis", *Proceedings of the Royal Society of London. Series A: mathematical, physical and engineering sciences*, *454*(1971), pp.903-995, 1998.

Huber, P. J. (1985). Projection pursuit. *The annals of Statistics*, 435-475, 1985.

Lambeck, K. (1980), The Earth's Variable Rotation: Geophysical Causes and Consequences, 449 pp., doi:10.1017/CBO9780511569579, Cambridge Univ. Press, New York.

Le Mouël, J.L., Lopes, F. and Courtillot, V., "A solar signature in many climate indices", *Journal of Geophysical Research: Atmospheres*, *124*(5), pp.2600-2619, 2019

Le Mouël, J. L., et al. "On forcings of length of day changes: From 9-day to 18.6-year oscillations." *Physics of the Earth and Planetary Interiors* 292 (2019): 1-11, 2019b.

Le Mouël, J. L., Lopes, F., & Courtillot, V. (2019). Singular spectral analysis of the aa and Dst geomagnetic indices. *Journal of Geophysical Research: Space Physics*, *124*(8), 6403-6417, 2019c

Le Mouël, J. L., F. Lopes, and V. Courtillot. "Solar turbulence from sunspot records." *Monthly Notices of the Royal Astronomical Society* 492.1: 1416-1420, 2020.

Lemmerling, P. and S. Van Huffel, "Analysis of the structured total least squares problem for hankel/toeplitz matrices", Numerical Algorithms 27 (1), 89–114. 162, 2001

Lockwood, M., "Long-term variations in the magnetic fields of the Sun and the heliosphere: Their origin, effects, and implications", *Journal of Geophysical Research: Space Physics*, *106*(A8), 16021-16038, 2001

Morlet, J., G. Arens, E. Fourgeau, et D. Glard, "Wave propagation and sampling theory—part i: Complex signal and scattering in multilayered media", Geophysics 47 (2), 203–221. 134, 1982.

Papoulis, A., "Probability, random variables and stochastic processes", McGraw-Hill. 80, 1984.

Vautard, R. et M. Ghil, "Singular spectrum analysis in nonlinear dynamics, with applications to paleoclimatic time series", Physica D-Nonlinear Phenomena 35, 395–424, 1989.

---

## Author Comment (AC1)

"A strong link between variations in sea-ice extent and global atmospheric pressure" by Le Mouël et al applies the singular spectrum analysis (SSA) method to both Arctic and Antarctic sea ice and sea-level pressure (SLP) time series to identify and compare common sets of harmonics between the respective time series. Further, temporal comparisons are made between sub-annual to multidecadal harmonics, and those of longer periodicity in the ice cover and SLP data are related to astronomical and astrophysical forcing cycles. A researcher with expertise in such cycles would be better equipped to evaluate and offer an opinion on the validity of such arguments in the context of both past and modern cryospheric change. However, that said, it is not apparent what the key, novel findings are from the study. My additional comments herein mainly encompass data and methodology concerns.

Seasonal change has a clear impact on ice growth and melt, but how climate change and related oceanic and atmospheric warming of the last two plus decades factor into the interpretation of these results is unclear.

As can be seen in Figures 04 to 09, for instance Figure 05, the forced (annual) oscillations are quite stable, both in amplitude and phase, from the end of the '70s to the Present. This stability is expected (*cf.* Lambeck, 2005, chapter 7, page 146, "*Seasonal variations*") since motion of the rotation pole (governed by the Liouville-Euler equations) involves periodic climatic excitations functions (as far as the annual period and its harmonics are concerned). The re-organization of masses at the Earth's surface and the annual oscillation are linearly related by this system of equations. We show in Figures A01a and A01b (attached to this reply) the annual components of the two coordinates of the rotation pole since 1846 (from Lopes et al., 2021). We see that modulation of the amplitudes is quite small and so is the phase difference between the two coordinates. This in itself is not a new result (it can be found in Figure 5.13, page 97 of Lambeck, 2005). The reviewer is right in that, contrary to the trends of our Figure 03, the effect of climate warming on this oscillation is not clear. But we believe that some of these results are interesting, in that they allow us to advance on the physical nature of the oscillating excitation functions that are generally included in the Liouville-Euler system of equations (chapter 4, page 47, Lambeck, 2005). Our main findings have to do with understanding and explicitly formulating the excitation functions in the Liouville-Euler equations. At the same time, we use the data to suggest that a physical mechanism could be found in the Taylor-Couette flow on a sphere (evidenced here through the harmonic suite of SSA

components at 1 year, 1/2, 1/3, 1/4 and 1/5 yr).

[Figure]

Figure A01a: Annual oscillations of the coordinates of the rotation pole (from Lopes et al., 2021)

[Figure]

Figure A01b: Phase difference between the annual components of the two coordinates of the rotation pole (from Lopes et al. 2021)

Further, to provide longer-term context to the results, the conclusions attempt to offer some insights between ice cover and SLP beyond the satellite era. This is problematic due to sparse data over the polar oceans (especially Southern Ocean) until the 1950s and thus likely impacts confidence in the sea ice and SLP periodicities calculated over that period, though no error estimates are provided accounting for this shortcoming. Below I outline more detailed concerns along these and editorial lines.

Indeed, as recalled above, there is a link between the re-organization of masses at the Earth surface and the mean pole of rotation. This link being a system of linear differential equations, it is legitimate to try (in the absence of data for sea ice) to move back in time using existing data.

Moreover, since here we are interested in oscillations with periods that are short compared to the time range of data, it is legitimate to predict (forward and backward) mechanisms that link them. In summary, the Liouville-Euler equations link the motion of the rotation pole and the mass re- organizations at the Earth surface. These linear partial differential equations (functions of time)

have been used extensively (e.g. Nakiboglu and Lambeck, 1980; Zotov et al., 2017). They provide a widely accepted physical model, which allows one to propose more general conclusions even with
incomplete data (see below), given that they be of short period compared to the data interval.

Regarding the impact on the components with annual and sub-annual periods that we
extract: The SSA method (*eg.* Vautard et Ghil, 1989; Golyandina & Zhigljavsky, 2013) allows one
to extract components with annual and sub-annual periods; we have checked and estimated more
precisely these periods with Fourier transforms. They are always accompanied by an associated
uncertainty (*cf.* Table 01). Since we discuss periodicities that are at least 42 times shorter than the
time range of the data (42 years) we are certain to comply with Shannon's criterion.

Besides, the sea ice time series appears to be stationary (stricto sensu) which is a necessary
condition to obtain a good Fourier transform (a bit more than two cycles would have been sufficient
to determine the annual oscillation). Following Claerbout (1976), 8 to 10 cycles would have been
sufficient in the case of a noisy stationary (senso latu) signal. We have 42!

The reviewer writes that we do not provide uncertainties or error estimates. As can be seen
in Table 01 such is not the case. It would certainly have been even better if the data themselves
(time series) had error estimates but such is unfortunately not the case.

*Specific Comments*

1) To reiterate, seasonal temperature and pressure changes due to annual earth-sun relations
certainly impact the annual cycle of ice growth and melt and presence and strength of
climatological pressure features. From SSA applied to sea ice and pressure time series, we would
expect related "cycles" to emerge at seasonal and annual scales through time. The rates of Earth's
air/ocean temperature changes, however, are not nearly as consistent through time.

We agree with this point of principle. But on this time scale our analysis does not show any
such variation. Let us recall that, regardless of the origins of the annual forcing, any modification
(in phase or amplitude) in the forcing must have an effect on polar motion (*cf.* Liouville-Euler). Yet,
as shown in Figures A01a and A01b, such does not seem to be the case. Ice does not form or melt
instantaneously, but integrates environmental conditions (including geophysical phenomena) over
time. The published literature reflects an ongoing debate regarding the climate and global warming,
in particular SLP: the rise in temperature must either affect the great atmospheric convection cells
(Hadley, Ferrel, etc …) or increase the number of extreme events (*e.g.* Chang , 1995; Dima and
Wallace, 2003; Frierson et al., 2007; Hu and Fu, 2007; Kharin et al., 2007; Lu et al., 2007; Tandon et al., 2013; Shepherd, 2014; Tao et al., 2016; Grise and Davis, 2020, Schaeffer et al., 2005; Stott et
al., 2010; Rahmstorf and Coumou, 2011; Rummukainen , 2012; Trenberth et al., 2015). As far as we
are concerned, only the (SSA) trends seem to undergo a strong amplitude modulation. This is more
in line with an impact on the longer periods rather than on periodicities as short as annual (*cf.*
Nakiboglu & Lambeck, 1980). However, this is not the purpose of the present study.

Is SSA an appropriate methodology to measure such evolving and covarying sea ice and
SLP change?

In principle, none of the methods of spectral analysis (be they Fourier transforms (FT),
wavelet transforms (WT) or SSA) is better than another one in an absolute sense. Again in principle,
the FT or WT can both be pushed to high orders to match the results we obtain with SSA. As an
example from our own group, results on polar motion obtained with wavelets by Gibert et al. (1998)
are identical with those of Lopes et al. (2021) using SSA. The SSA method is particularly well
explained in Golyandina  & Zhigljavsky (2013); it originates from the scientific fields of climate
and paleoclimate (before being put on a complete mathematical basis, note that the Cooley-Tuckey
method for Fourier transforms or the wavelet transform were "invented" by geophysicists). Here is
an incomplete list of (productive) applications of SSA:  Vautard and Ghil  (1989), Vautard et al.
(1992), Zhaomin & Shisong (1996), Jevrejeva and Moore (2001), Kravtsov and Ghil (2004), Darby
et al. (2017), Pepelyshev & Zhigljavsky (2017). All of these papers deal with climate, sea ice,
and/or temperature: it is because of these successful studies that we have selected to apply SSA in
this paper.

How does global climate change and related oceanic and atmospheric warming exacerbated
at both poles during at least the last two decades (i.e., Arctic amplification) factor into the
interpretation of your results and the purported astronomical and astrophysical forcings linked with
non-stationary sea ice and perhaps SLP behaviors?

As explained in our section 5.3 on mechanisms, our aim is to try to understand and to
explicitly formulate the excitation functions in the Liouville-Euler equations. At the same time, we
use the data to suggest that a physical mechanism could be found in the Taylor-Couette flow on a
sphere, which we believe is an important problem; moreover, it is evidenced here through the
harmonic suite of SSA components at 1 year, 1/2, 1/3, 1/4 and 1/5 yr. As stated above, we do not
know whether global warming impacts the annual oscillations; there is more information in the trends of Figure 03 that vary a lot. We have confirmed a result that has been obtained previously by others, clearly seen in the raw data of Figure 01: the annual variations have remained stable for 42 years and have not been affected by global warming on this time scale. If they had been, we would not have found the harmonic suite of components.

2) For satellite-era comparisons against sea ice variability, why use the coarse resolution HadSLP2 and not a newer, higher spatiotemporal resolution product such ERA5? There is quite a difference in spatial resolution between these two products and ERA5 assimilates lots of new data sources. At minimum, more justification for HadSLP2 over a newer product like ERA5 needs to be provided. The data quality/quantity issue further plays into the longer-term interpretation of results mentioned in the following comment.

For two reasons. We are interested in understanding global scale phenomena such as polar motion. Once again, there is a link (through the Liouville-Euler equations) between re-organization of surface masses and polar rotation. The latter is a scalar, available with a rough sampling of 1 month since 1840. For coherency, all our analyses are performed on data/time series with the same (rough) sampling. The second reason is that the data set HadSLP2 is to our knowledge the most widely used, studied and published. We were not aware of the ERA5 data set and thank the reviewer for pointing it out to us. We certainly intend to use it in a future study.

3) In providing long-term context to the core study results, the conclusions need to be modified. Meteorological data including surface pressure is very sparse for the Southern Ocean and Antarctica, especially prior to the IGY (~1957-1958). This data quantity issue is recognized in the conclusions of Allan and Ansell (2006), which provides an overview of the HadSLP2 dataset used in the paper. Further, many gridded products have questionable data quality with a scarce number of observations included before the first half of the twentieth century (Fogt et al., 2018 J. Climate). How sparse are Southern Ocean data observations comprising the HadSLP product pre-dating the satellite era, let alone during this era from which the main results are built? Much like the passive microwave ice cover record, a description of the HadSLP dataset construction and available observations through the pressure record need to be discussed and results emphasized for periods when the data quality/quantity are most robust. These dataset issues need to be kept in perspective when interpreting the results back beyond the IGY and to the 1840s.

We agree with the reviewer and remove Figure 11 and all comments linked to it.

*Technical Corrections*

Some editorial remarks and clarifications are listed by line number (L):

L17: "It fits topographic forcing." – what does? Please clarify.

This anomaly is seen in maps of correlations of variations in sea-ice extent with atmospheric pres-
sure, surface temperature and winds. It fits the regional topography.

L44: Change "identifications" to "identification"

OK done

L48: Spell out the climate indices (e.g., AO, AAO) where first introduced.

Arctic Oscillation, Antarctic Oscillation.

L51-52: The AO is commonly a statistical solution based on a univariate geopotential height field
(e.g., NOAA CPC uses the 1000 hPa GPH field). Please clarify this description.

Aren't AO and AAO EOFs rather than statistical solutions? Do we need to restate the definition of
AO and AAO?

L63: Remove "lod"

OK done

L67: This sentence is confusing. Please re-write to clarify its intent.

We have applied SSA to study the variations and oscillatory components of a number of solar, cli-
matological and geophysical phenomena, parameters and proxies (see the above references), but not
yet atmospheric pressure (AP).

L84: "quoting Cavalieri et al" with what? Methods? Please clarify what is meant here.

After 2010 and up to 2018, we refer to Parkinson (2019).

L116: The Allan and Ansell paper was published in 2006 not 2004.

OK corrected.

L120: In addition to specific comments above, references to previous studies that have used the data
for polar research would help support the statement.

We found it very difficult to check all aspects of this complex procedure and prefer, as in most of our previous studies of similar series, to trust the available data.

Figure 2 (and others): Y axis labels referencing surface pressure should consistently list "hPa."

Check that these pressure units are consistently referenced through the paper.

hP changed to hPa on the Oy axis of Figures 2,3,4,6,7,8.

L297: Reference and description of the earth-sun distance data should be provided in the data section.

OK done

REFERENCES

Chang, E. K., (1995) "The influence of Hadley circulation intensity changes on extratropical climate in an idealized model", *Journal of Atmospheric Sciences*, *52*(11), 2006-2024.

Claerbout, J. F. (1976), "*Fundamentals of geophysical data processing*", (Vol. 274). McGraw-Hill, New York.

Darby, D. A., Andrews, J. T., Belt, S. T., Jennings, A. E., & Cabedo-Sanz, P. (2017), "Holocene cyclic records of ice-rafted debris and sea ice variations on the East Greenland and Northwest Iceland margins", *Arctic, Antarctic, and Alpine Research*, *49*(4), 649-672.

Dima, I. M. and Wallace, J. M., "On the seasonality of the Hadley cell", *Journal of the atmospheric sciences*, *60*(12), 1522-1527, 2003.

Frierson, D. M., Lu, J., & Chen, G. (2007), "Width of the Hadley cell in simple and comprehensive general circulation models", *Geophysical Research Letters*, *34*(18).

Gibert, D., Holschneider, M., & Le Mouël, J. L. (1998), "Wavelet analysis of the Chandler wobble", *Journal of Geophysical Research: Solid Earth*, *103*(B11), 27069-27089.

Grise, K. M. and Davis, S. M., (2020) "Hadley cell expansion in CMIP6 models", *Atmospheric Chemistry and Physics*, *20*(9), 5249-5268.

Golyandina, N., & Zhigljavsky, A. (2013), "*Singular Spectrum Analysis for time series*", (Vol. 120). Berlin: Springer.

Hu, Y. and Fu, Q. (2007), "Observed poleward expansion of the Hadley circulation since 1979", *Atmospheric Chemistry and Physics*, *7*(19), 5229-5236.

Kharin, V. V., Zwiers, F. W., Zhang, X. and Hegerl, G. C., (2007) "Changes in temperature and precipitation extremes in the IPCC ensemble of global coupled model simulations", *Journal of Climate*, *20*(8), 1419-1444.

Kravtsov, S., and M. Ghil., (2004), "Interdecadal variability in a hybrid coupled ocean–atmosphere–sea ice model", *Journal of physical oceanography*, 34.7 (2004): 1756-1775.

Jevrejeva, S., & Moore, J. C. (2001), "Singular spectrum analysis of Baltic Sea ice conditions and large-scale atmospheric patterns since 1708", *Geophysical Research Letters*, *28*(23), 4503-4506.

Lambeck, K. (2005), "*The Earth's variable rotation: geophysical causes and consequences*", Cambridge University Press.

Lopes, F., Le Mouël, J. L., Courtillot, V., & Gibert, D. (2021), "On the shoulders of Laplace", *Physics of the Earth and Planetary Interiors*, *316*, 106693.

Lu, J., Vecchi, G. A. and Reichler, T., (2007), "Expansion of the Hadley cell under global
warming", *Geophysical Research Letters*, *34*(6).

Nakiboglu, S. M., and Lambeck, K. (1980). "Deglaciation effects on the rotation of the
Earth", *Geophysical Journal International* 62.1: 49-58.

Pepelyshev, A., & Zhigljavsky, A., (2017), "SSA analysis and forecasting of records for
Earth temperature and ice extents", *Statistics and Its Interface*, *10*(1), 151-163.

Rahmstorf, S., and Coumou, D., (2011) "Increase of extreme events in a warming world",
*Proceedings of the National Academy of Sciences*, *108*(44), 17905-17909.

Rummukainen, M., (2012) "Changes in climate and weather extremes in the 21st century",
*Wiley Interdisciplinary Reviews: Climate Change*, *3*(2), 115-129.

Schaeffer, M., Selten, F. M., & Opsteegh, J. D., (2005) "Shifts of means are not a proxy for
changes in extreme winter temperatures in climate projections", *Climate Dynamics*, *25*(1), 51-63.

Shepherd, T. G., (2014), "Atmospheric circulation as a source of uncertainty in climate
change projections", *Nature Geoscience*, *7*(10), 703-708.

Stott, P. A., Gillett, N. P., Hegerl, G. C., Karoly, D. J., Stone, D. A., Zhang, X., & Zwiers, F.,
(2010), "Detection and attribution of climate change: a regional perspective", *Wiley*
*Interdisciplinary Reviews: Climate Change*, *1*(2), 192-211.

Tandon, N. F., Gerber, E. P., Sobel, A. H., and Polvani, L. M., "Understanding Hadley cell
expansion versus contraction: Insights from simplified models and implications for recent
observations", *Journal of climate*, *26*(12), 4304-4321, 2013.

Tao, L., Hu, Y., & Liu, J., (2016) "Anthropogenic forcing on the Hadley circulation in CMIP5
simulations", *Climate dynamics*, *46*(9-10), 3337-3350.

Trenberth, K. E., Fasullo, J. T., and Shepherd, T. G., (2015), "Attribution of climate extreme
events", *Nature Climate Change*, *5*(8), 725-730.

Vautard, R., & Ghil, M. (1989), "Singular spectrum analysis in nonlinear dynamics, with
applications to paleoclimatic time series", *Physica D: Nonlinear Phenomena*, *35*(3), 395-424.

Vautard, R., Yiou, P., & Ghil, M. (1992). Singular-spectrum analysis: A toolkit for short,
noisy chaotic signals. *Physica D: Nonlinear Phenomena*, *58*(1-4), 95-126.

Zhaomin, W., & Shisong, H. (1996), "The spatial and temporal variation characteritics of
arctic and antarctic sea ice", *Scientia Meteorologica Sinica*, *4*.

Zotov, Leonid, et al., (2017), "Multichannel singular spectrum analysis of the axial
atmospheric angular momentum", *Geodesy and Geodynamics*, 8.6 (2017): 433-442.